# Molecular Mechanisms of *Borrelia* *burgdorferi* Phagocytosis and Intracellular Processing by Human Macrophages

**DOI:** 10.3390/biology10070567

**Published:** 2021-06-22

**Authors:** Philipp Woitzik, Stefan Linder

**Affiliations:** Institute for Medical Microbiology, Virology and Hygiene, University Medical Center Eppendorf, 20246 Hamburg, Germany; philipp.woitzik@stud.uke.uni-hamburg.de

**Keywords:** Borrelia, coiling pseudopod, filopodia, Lyme disease, macrophages, phagosome, phagocytosis

## Abstract

**Simple Summary:**

Borreliae are a group of highly motile bacteria that are characterized by their corkscrew-like shape. They can be transferred by a tick bite to a human host and cause severe illnesses. Accordingly, an untreated infection with *Borrelia burgdorferi* can lead to the development of Lyme disease, which can affect the skin, joints, heart and nervous system. It is thus important to understand how borreliae interact with the human immune system, and which mechanisms lead to their depletion in the human body. Macrophages are part of the innate immune system and among the first cells that encounter invading borreliae. In this review, we discuss the molecular mechanisms that enable macrophages to recognize, take up and digest borreliae. We also point out potential ways in which borreliae might evade these mechanisms.

**Abstract:**

Lyme disease is the most common vector-borne illness in North America and Europe. Its causative agents are spirochetes of the *Borrelia* *burgdorferi* *sensu* *latu* complex. Infection with borreliae can manifest in different tissues, most commonly in the skin and joints, but in severe cases also in the nervous systems and the heart. The immune response of the host is a crucial factor for preventing the development or progression of Lyme disease. Macrophages are part of the innate immune system and thus one of the first cells to encounter infecting borreliae. As professional phagocytes, they are capable of recognition, uptake, intracellular processing and final elimination of borreliae. This sequence of events involves the initial capture and internalization by actin-rich cellular protrusions, filopodia and coiling pseudopods. Uptake into phagosomes is followed by compaction of the elongated spirochetes and degradation in mature phagolysosomes. In this review, we discuss the current knowledge about the processes and molecular mechanisms involved in recognition, capturing, uptake and intracellular processing of *Borrelia* by human macrophages. Moreover, we highlight interactions between macrophages and other cells of the immune system during these processes and point out open questions in the intracellular processing of borreliae, which include potential escape strategies of *Borrelia*.

## 1. Introduction

Lyme disease or Lyme borreliosis is a multisystemic bacterial infection, manifesting primarily in the skin, the nervous system and joints [1]. It is the most common vector-borne illness in North America and Europe [2]. Lyme disease was first described as an epidemic form of juvenile arthritis in Lyme, Connecticut in 1976 [3]. In 1982, Willy Burgdorfer identified a spirochetal bacterium isolated from ticks of the Ixodes family, and from patient samples, as the causative agent. The spirochete was accordingly called *Borrelia burgdorferi* [4]. Borreliae are morphologically characterized as irregularly coiled spirochetes that are 10–40 µm in length and up to 0.3 µm in diameter [5]. They are highly motile, which is based on the presence of periplasmic flagella, and can reach velocities of up to 4.25 µm/s [6]. To date, 52 species of *Borrelia* have been identified, 21 of which are associated with Lyme disease [7]. Borreliae are mostly found in rodents and birds, and transmitted by tick bites, with infection of humans happening inadvertently [8]. In Europe and Asia, *Borrelia afzelii*, *Borrelia garinii*, *Borrelia burgdorferi* and *Borrelia bavariensis* are known to cause Lyme disease [9,10], while in North America, it is primarily caused by *Borrelia burgdorferi* [11]. New pathogenic strains are apparently emerging in Canada and the northern United States [12]. The group of *Borrelia* which cause Lyme disease is collectively referred to as the *Borrelia burgdorferi sensu latu* complex, whereas *Borrelia burgdorferi* as a species is addressed as *Borrelia burgdorferi sensu stricto* [10].

The course of Lyme disease can be divided into three distinct stages—the early localized infection, the early disseminated infection and late-stage Lyme disease—though the infection is not necessarily clinically apparent during all three stages [13]. In over 50% of cases, patients develop a painless skin rash called erythema migrans, which spreads in a characteristic ring-like morphology from the site of the tick bite 7–10 days after the infection [1]. Erythema migrans develops as a sign of the localized immune response and was shown to contain cells of the innate immune system, such as macrophages, neutrophils and dendritic cells [14]. *Borrelia* can enter the vasculature during the tick-bite and escape from the vascular system in a multi-step process comprising tethering, dragging, stationary adhesion and extravasation, leading to the distribution of the pathogen throughout the body [15]. Depending on the genospecies of the infections, roughly 5–60% of untreated erythema migrans progress into systemic infections, which can be divided into early disseminated infections and late-stage Lyme disease, according to the duration of symptom persistence [1,13]. Of note, ~50% of patients in the disseminated stages of the infection report no prior erythema migrans [1]. Common manifestations include arthritis, carditis and neuroborreliosis, with variations in frequency depending on the underlying genotype of the infection [13]. In some cases, subjective symptoms such as fatigue, cognitive difficulties, myalgia and arthralgia persist for a significant period of time even after antibiotic treatment. The underlying etiology of the so-called post(treatment)-Lyme disease syndrome is not fully understood. Possible explanations range from microbial persistence or immune dysregulation to altered neural networks [16,17].

The immune response of the host is an important factor for preventing development or progression of Lyme disease. Of note, skin biopsies from erythema migrans are enriched in macrophages, neutrophils and dendritic cells [14]. These cells are part of the innate immune system and are thus the first line of defense against the infection. Here, we focus on the role of macrophages in the elimination and potential persistence of *Borrelia* in the human host.

To successfully eliminate borreliae, macrophages have to perform a series of finely tuned processes (Figure 1). At first, the immune cells need to recognize, capture and immobilize the highly motile *Borrelia* cells [18]. To achieve this, macrophages initially form rigid actin-rich protrusions called filopodia that contain closely-bundled, unbranched actin filaments [19,20]. Subsequently, a more flexible actin-rich structure, the coiling pseudopod, wraps around the captured bacteria and pulls them into the phagosome, a special compartment derived from the cell membrane. The early phagosome then matures progressively into the late phagosome and the phagolysosome. This process is accompanied by the compaction of the elongated spirochete [21], by progressive acidification of the compartment and the acquisition of lytic enzymes. These steps and the respective molecular mechanisms will be discussed in the following section. Please note that the term “*Borrelia*” in this review refers to members of the *Borrelia burgdorferi sensu latu* complex, especially in the context of Lyme disease, and not to other species such as *B. recurrentis* or *B. miyamotoi* that cause other infectious diseases. It is likely that several of the described mechanisms also apply to these other *Borrelia* species, but this has not been formally proven yet.

## 2. Phagocytic Uptake of *Borrelia burgdorferi* by Macrophages

Phagocytosis is defined as a multi-step process comprising recognition and capture of a particle larger than 0.5 µm in diameter, its internalization and its subsequent degradation [22]. It is an important factor in host defense against infection and a significant part of the immune (and inflammatory) response [23,24]. In mammals, some cells, including macrophages, dendritic cells, osteoclasts and eosinophils, show high phagocytic activity and efficiency and are therefore called “professional phagocytes” [25,26].

Phagocytes can recognize *Borrelia* through a set of receptors, including the Fcγ-receptor (FcγR), which recognizes opsonized borreliae through the constant fraction of bound immunoglobulins [27,28,29,30], and the complement receptor-3, which facilitates uptake through recognition of two of the major lipoprotein components of the spirochete’s cell membrane, outer surface proteins A and B (OspA and OspB), in an iC3b-independent manner [31,32,33]. Detection of other spirochetal lipoproteins or pathogen associated molecular patterns (PAMPs) through pattern recognition receptors such as Toll-like receptors 2 (TLR2) and 3 (TLR3) has also been shown to be part of the recognition [34,35,36,37]. The downstream signaling of TLR2 and TLR3 proceeds through myeloid differentiation factor 88 (MyD88)-dependent and MyD88-independent pathways, involving, among others, phosphoinositide 3-kinase (PI3K), an important trigger of actin polymerization [37,38]. Deficiency in TLR2, CR3 or FcγR leads to increased severity of symptoms and decreased numbers of neutralized *Borrelia* [39,40,41]. Of note, macrophages can adhere to both opsonized and non-opsonized *Borrelia*, but opsonization of the bacteria through *Borrelia*-specific antibodies or through serum, which contains factors of the complement system, has been shown to increase adhesion rates by a factor of 4–5 [42]. A series of other receptors has been detected by mass spectrometry in *Borrelia*-containing phagosomes, and shRNA-mediated depletion showed that some of them, including PLAUR (uPAR), CLEC13A, CLEC4N (Dectin 2), CLEC4B1, MARCO, STABILIN2 and CD33, are important for phagocytic uptake, whereas others, such as LY6E, CD59A, CD24A, CLEC4D, CLEC10A, STABILIN1, MSR1 and SIGLEC5, seem to negatively regulate phagocytosis [30]. Collectively, this evidence suggests that multiple receptors and signaling pathways act synergistically to facilitate efficient recognition and uptake of *Borrelia* by immune cells.

To increase their chances of encountering a pathogen, activated macrophages probe their environment by using actin-based cell protrusions called filopodia that are also enriched in receptor proteins (Figure 2) [43]. Co-incubation of primary human macrophages with live or heat-killed *Borrelia burgdorferi* greatly increased the number and the length of filopodia per cell, whereas incubation with supernatant from *Borrelia* cultures did not induce any noticeable changes, indicating that this process is actively driven by the macrophage upon encountering the spirochete [20]. Filopodia are elongated, finger-like structures that contain bundles of F-actin and undergo constant extension and retraction [43,44,45], based on actin polymerization dynamics. Consequently, members of the formin family, which are regulators of linear actin filaments, have been shown to be crucial for these processes. Formins are involved in the elongation, nucleation, bundling and severing of actin filaments, and the actual activity is dependent on the specific isoform [46]. For filopodia formation in macrophages, three formins are of major importance: mDia1, FMNL1 and Daam1. mDia1 localizes at the tip of the protrusion and is correspondingly important for actin filament elongation [20,47]. FMNL1, localized both at the tip and the shaft of filopodia, exhibits actin severing activity that creates barbed ends, which supports the growth of new filaments [20,48]; and Daam1, which is enriched at the shaft, is involved in actin bundling, thereby contributing to the stability of the protrusion [19,20]. In addition, fascin, another actin bundling protein, is also located along the shaft of the filopodia [19]. SiRNA-mediated depletion of either of these proteins led to significant decreases in the number of filopodia contacting *Borrelia* and in the uptake of *Borrelia* into macrophages [19,20]. Generally, a basal branched actin-network created by Arp2/3 [49] is required as the foundation for a subset of filopodia [50], although its specific requirement for the formation of *Borrelia*-induced filopodia has not been evaluated so far. To prevent the encountered pathogen from escaping, filopodia are able to pull it towards the cell surface or at least immobilize it [43,51,52,53], which is especially important when macrophages are confronted with highly motile pathogens, such as *Borrelia*. Inhibition of actin turnover using jasplakinolide led to a loss of this ability to exert a force on the captured pathogen [43].

Subsequent to capture by filopodia, borreliae are internalized by the formation of uptake structures that are also based on actin dynamics. For macrophages, two modes of internalization have been demonstrated: (1) Conventional phagocytosis, a process in which multiple cell protrusions form symmetrically around the attached pathogen and enwrap it. (2) A specialized process called coiling phagocytosis [54,55], which seems to be the predominant pathway for *Borrelia* uptake [54]. In this process, a single major actin-rich protrusion extends along the spirochete, closely following its helical morphology, and wraps itself multiple times around the elongated pathogen (Figure 2). Novel data suggest that in addition to the predominant coiling pseudopod, multiple small membrane folds support this process by attaching themselves to the pathogen at several places along its length [56]. Similarly to immobilization by filopodia, this process is actively driven by the host cell, as both live and heat- or chemically-killed *Borrelia* were observed to be taken up by this process. In contrast, neither supernatant from bacterial culture nor fragmented bacteria induced uptake by coiling phagocytosis, suggesting that this uptake mechanism is a specific response to the elongated helical morphology of the spirochete [57]. Interestingly, formation of these pseudopodia is linked to Daam1, one of the formins also responsible for the formation of filopodia, [19], which makes Daam1 the only actin regulator with demonstrated activity at both actin-based uptake structures for borreliae so far. In addition to Daam1, which is localized along the entire length of the pseudopod, the Arp2/3 complex is found in the form of dot-like accumulations at the turning points of the coiling pseudopod [18,42]. As Arp2/3 creates branched actin networks [49], the current hypothesis is that these Arp2/3-based spots could create “hinges” along the pseudopod, which, in combination with the more rigid, unbranched actin bundles created by Daam1, confer the necessary flexibility for the formation of the helical pseudopod [18]. The small GTPases CDC42 and Rac1, members of the Rho family, are upstream regulators of Arp2/3. Consistently, inhibition of CDC42 and Rac1 via microinjection of constitutively inactive mutants or C3-transferase, a Rho inhibitor, led to decreased formation of coiling pseudopods [42]. In addition, this pathway involves activation of the Arp2/3 complex by the nucleation-promoting factor WASP (Wiskott–Aldrich syndrome protein) [42,58]. As previously mentioned, downstream signaling of TLR2 and TLR3 involves MyD88-dependent and independent pathways. Recent data from MyD88-deficient murine bone-marrow derived macrophages (BMDMs) showed no change in adhesion to *Borrelia* and no deficiency in phagosomal degradation, but reduced uptake of borreliae. Furthermore, MyD88-signaling upregulates the expression of Daam1, Rac1, CDC42 and Akt1. Collectively, these data suggest a prominent role for MyD88 in regulating the formation of coiling pseudopods [37].

Following enwrapment by the coiling pseudopod, the entire phagocytic complex consisting of the pathogen and the pseudopod is internalized by a series of fusion events between adjacent membrane parts, thereby creating an enclosed compartment which subsequently pinches off from the plasma membrane [54]. Of note, owing to the pronounced length of the spirochete, internalization can already begin while substantial parts of the bacterium are still extracellular. It may thus take longer than 40 min for a single spirochete to be completely internalized [18].

## 3. Intracellular Processing of *Borrelia burgdorferi* in Macrophages

As mentioned, intracellular processing of a *Borrelia* cell can start even before it is internalized in its entirety [21,56]. Generally, internalized bacteria are taken up into a phagosome, which constitutes a plasma membrane-derived compartment that is initially filled with fluids from the extracellular space [59]. In the case of *Borrelia* internalization, nascent phagosomes are still accessible from the extracellular space. Of note, in nascent phagosomes, the phagosomal membrane only loosely follows the form of the spirochete, resulting in infrequent points of contact between the membrane and the pathogen [56]. Collectively, these areas of close contact form an ever-increasing barrier to the extracellular space. In consequence, *Borrelia*-specific antibodies could only reach proximal areas within the phagosome [56], although experiments using the cell surface markers concavalin A-gold, ruthenium red and tannic acid showed that the internal membranes of the nascent phagosomes could be stained up to the time of complete enclosure of the spirochete, indicating that fluids and small solutes could still access the entire length of the unclosed phagosome [54].

In approximately 2–5% of the cases, the phagosomal membrane enwrapping the internalized part of the spirochete was shown to extend into the host cytoplasm beyond the tip of the pathogen for more than 10 µm. These structures, called tunnels, might point to partial escape of the highly motile *Borrelia* from the unclosed phagosome (Figure 1). In macrophages transfected with RFP-LactC2, a reporter of phosphatidylserine and therefore the outsides of early phagosomes, the signal confirmed that these tunnels are indeed part of the phagosome and are derived from the plasma membrane [56]. Collectively, this demonstrates that uptake of *Borrelia* by macrophages is not a simple, one-directional process but rather an active “tug-of-war” between highly motile borreliae and the phagocyte.

### 3.1. Compaction of Borrelia in the Phagosome

An early step during the intracellular processing of *Borrelia* is the compaction of the elongated spirochete into a globular form, which is associated with a reduction in phagosomal surface (Figure 3) [21]. This reduction of the membrane surface area is likely due to the extrusion of membrane tubules, which could be observed frequently, especially at sites of curvature discontinuity [21,56]. Multiple proteins have been identified which play crucial roles in the regulation of this process, including RabGTPases, sorting nexin 3 (SNX3) and galectin-9 [60]. RabGTPases are regulators of vesicle trafficking and vesicle fusion with and fission from membranes [61]; sorting nexins are a family of proteins involved in intracellular trafficking and possess a phospholipid binding PX domain [62]; the galectin family is involved in carbohydrate binding on the cell surface, but also in the regulation of intracellular trafficking pathways [63].

Initially, *Borrelia* are internalized in a Rab22a-positive phagosome, which is in turn fused with vesicles that are positive for both Rab5a and SNX3 (Figure 3). siRNA-mediated depletion of either of those proteins significantly decreased the proportion of compacted *Borrelia* [21,60]. Interestingly, these points of contact between phagosome and vesicles are especially found at sites of altered membrane curvature on the phagosomal membrane, which also constitute the sites of membrane tubule formation and abscission [18,21,60]. It has been shown that Rab5a vesicles contact the phagosomal coat through binding of vesicle-localized SNX3 with the phagosomal phospholipid PI(3)P (Phosphatidylinositol 3-phosphate) [60]. In particular, a SNX3 construct carrying a point mutation in the PX domain and thus being unable to bind PI(3)P, was unable to rescue compaction in SNX3-depleted macrophages, and inhibition of PI(3)P formation using the PI3K class III inhibitor wortmannin significantly decreased compaction rates as well. Collectively, these data lead to the conclusion that the interaction between SNX3 and PI(3)P is necessary for the compaction of *Borrelia* within phagosomes [60]. Interestingly, PI3 kinase (PI3K), one of the enzymes able to generate PI(3)P, has been shown to be particularly active at sites of altered membrane curvature [64], and such sites are a natural consequence of the helical spirochete morphology. Of note, PI(3)P was shown to be gradually enriched at the phagosome. This enrichment was accompanied by only occasional contact with PI(3)P-positive endocytic vesicles, suggesting local generation of the phospholipid at the phagosome surface, with an only accessory influx of vesicle-delivered PI(3)P [60].

It is thus likely that the helical shape of *Borrelia* and the resulting phagosome morphology lead to increased PI3K activity at helical turns, thereby generating docking sites for Rab5a vesicles through a SNX3–PI(3)P interaction [60]. Moreover, SNX3 does not only act as an adaptor for Rab5a vesicles, as it also binds galectin-9. Consistently, a SNX3 construct lacking its C-terminal region, an 11 amino acid residue stretch identified to be responsible for the interaction with galectin-9, failed to rescue regular levels of compaction in SNX3 knockdown cells. Interestingly, galectin-9 is present at a vesicle population that is also positive for flotillin-2 [60], a protein implicated in trafficking, signal transduction and endocytosis [65]. Furthermore, this vesicle population is distinct from the SNX3/Rab5a vesicle population, and is only recruited to borreliae phagosomes after SNX3/Rab5a vesicle docking. However, siRNA-mediated depletion of galectin-9 leads to a comparable reduction in phagosomal compaction. Depletion of both galectin-9 and SNX3 simultaneously did not result in additive effects, indicating that both proteins act in the same pathway that regulates compaction of the *Borrelia*-containing phagosome [21,60]. Interestingly, macrophages depleted in Rab22a, Rab5a, SNX3 or galectin-9, either singly or in combination, were still able to maintain ~50% of regular phagosome compaction levels, hinting at the existence of one or more alternative pathways for this process [21,60].

Importantly, in addition to the phagosome and its associated vesicle populations, the endoplasmic reticulum (ER) has emerged as a major regulator of *Borrelia* intracellular processing. Multiple ER tubules were observed in close proximity to phagosomes [21] which were nascent, incompletely closed or in the process of compaction (Figure 3) [56]. Recent data also demonstrated the presence of multiple bona fide ER contact sites at *Borrelia*-containing phagosomes that were positive for the marker STIM1 [56]. Moreover, SNX3/Rab5a vesicles have been shown to be tethered to the ER [21,56], comparably to what was previously shown for endosome-ER interactions [66]. It is thus very likely that ER contacts with vesicles and phagosomes play both structural and functional roles in the maturation process of the *Borrelia*-containing phagosome, which will also be discussed in the next section.

### 3.2. Phagosomal Maturation and Development of the Phagolysosome

Generally, the early phagosome is still filled with fluid derived from the extracellular space and does not show any bactericidal activity. Shortly after the separation from the cell membrane and the sealing of the phagosome, further maturation begins and endocytic vesicles fuse with the phagosome, drastically changing the composition of its contents, leading to acidification and the acquisition of enzymes necessary for the degradation of the pathogen. This new hybrid organelle is called the phagolysosome [59,67]. As discussed, phagosomal compaction of initially elongated *Borrelia* cells seems to be an important step in the maturation of the phagosome, as depletion of many of the regulators involved in this step also results in defects in the further maturation process.

RabGTPases, a family of intracellular trafficking regulators [61], are involved in both steps of borreliae intracellular processing. Generally, individual Rab family members show distinct localizations within the cell, often in specific membranes or compartments, such as the phagosome. They are involved in regulating the compositions of these membranes; in the movement of organelles by regulating the interactions with elements of the cytoskeleton [68]; in membrane fusion and fission events; and in the trafficking of vesicles [61]. As discussed above, Rab22a was found to be enriched at the coat around the early *Borrelia*-containing phagosome; vesicles positive for Rab5a contact the Rab22a coat especially at sites of altered membrane curvature (Figure 4) [21]. Both proteins play an important role in the compaction of the elongated phagosome into a globular structure, but also in the maturation of the phagosome into an organelle capable of degrading the pathogen, as siRNA-mediated knockdown of either protein led to a significant decrease in proteolytic activity of *Borrelia*-containing phagosomes, as determined by reduced fluorescence of DQ-BSA (dequenched bovine serum albumin), a reporter of proteolysis. Consequently, internalized borreliae show higher survival rates in cells depleted in Rab22a or Rab5a [21,60].

How can the molecular interplay between Rab5a and Rab22a in borreliae phagosome maturation be envisioned? Generally, Rab5 and its effector EEA1 (early endosomal antigen 1) have been described as important regulators for the trafficking of early endosomes [69], and another RabGTPase, Rab7, was found to be crucial for the development of the late endosome and endosomal degradation [70]. Rab5a and Rab7 have also been observed in early and late phagosomes, respectively. As phagosomes are known to interact with the endocytic pathway, it seems likely that they play similar roles in phagosome maturation [59,67,71]. For other pathogens, especially *Mycobacteria*, a crucial role for Rab22a was found in the conversion from Rab5-positive to Rab7-positive phagosomes, and a lack of this conversion led to an arrest of phagosome maturation [71]. Considering the detection of Rab7 in *Borrelia*-containing phagosomes and also the impaired degradation of *Borrelia* in Rab22a-deficient macrophages [21], a similar role for Rab22a in the conversion from Rab5a-positive to Rab7-positive phagosomes could be possible. Moreover, the involvement of other Rab proteins is also likely, as several Rabs have been linked to phagosome maturation [72] and have also been detected at *Borrelia* phagosomes [21], although their potential impact on maturation of *Borrelia*-containing phagosomes is currently unclear. This should be fertile ground for future research.

The acquisition of endosomal enzymes by phagosomes is thought to happen through a series of transient contacts with endosomal compartments in a “kiss-and-run” fashion (Figure 4) [73]. These fusion events between the respective stages of phagosomes and endosomes are facilitated by the RabGTPases that are predominant in both compartments at these stages [73,74]. This general concept likely also applies for *Borrelia* phagosome maturation, as both Rab5a/SNX3-positive and galectin-9-positive vesicles were observed to repeatedly contact borreliae-containing phagosomes, but not fuse with them [21,60]. Generally, in addition to RabGTPases, a role for SNAREs (soluble N-Ethylmaleimide-sensitive factor-attachment protein receptors), and their binding partners, SNAPs (NSF-attachment proteins), has been shown for the targeting of vesicles in the endosomal and phagosomal progression [59,75]. However, data specifically for borreliae phagosomes are currently lacking. A direct interaction between EEA1, a downstream effector of Rab5, and syntaxin 13, a SNARE protein, has been described to be necessary for early membrane fusion events in a cell-free system of endosome fusion [76,77]. It is thus possible that, in addition to the previously identified Rab proteins, SNARE complexes could be involved in the regulation of these interactions between the endosomal and phagosomal pathways in *Borrelia* phagosome progression.

WASH (Wiskott–Aldrich syndrome protein and SCAR homologue) is a known regulator of the Arp2/3 complex and thus of actin polymerization [78]. In *Dictyostelium*, a eukaryotic bacterivore, loss of WASH did not alter the rate of phagocytosis of latex beads, but led to reduced proteolysis within the phagosomes and reduced acquisition of lysosomal enzymes [79]. As phagocytosis is a highly conserved process between lower and higher eukaryotic cells [80] and fusion events between late endosomes and latex bead-containing phagosomes depend on F-actin polymerization [81], a role of WASH in the interaction between the phagocytic and endocytic pathway during *Borrelia* processing seems possible.

During the process of maturation, phagosomes generally establish progressively more acidic conditions inside, a step thought to be necessary for the bactericidal ability, as (1) most lysosomal enzymes prefer these acidic conditions, (2) the low pH supports denaturation and (3) the creation of reactive oxygen species requires protons [59,82,83]. These protons are actively transported into the phagosome by the vacuolar-type proton transporting ATPase (V-ATPase) [59], which is in general recruited in the early stages of the phagosome and colocalizes with Rab7, a marker progressively enriched during the maturation of endosomes and phagosomes [82,83]. In murine macrophages, depletion of the V-ATPase disrupted acidification of latex bead-containing phagosomes and impaired the bactericidal ability of *E. coli* phagosomes, yet the recruitment of lysosomal enzymes was unhindered [84]. Indeed, inhibition of the V-ATPase using the inhibitor bafilomycin also resulted in a loss of acidification of *Borrelia*-containing phagosomes [21], pointing to the significance of this enzyme in the context of intracellular degradation of *Borrelia* by macrophages (Figure 4).

Hydrolases constitute another group of enzymes that are usually acquired by phagosomes through the mentioned fusion events with lysosomes. More than 50 hydrolases have been identified within lysosomes, comprising mostly cysteine, aspartic and serine proteases that are predominantly members of the cathepsin family [85]. Initial data showed colocalization between cathepsin L and *Borrelia*-containing phagosomes in murine macrophages [86]. However, it is not yet known which hydrolases are indeed required for proteolytic processing of *Borrelia* in phagolysosomes. In addition, the phagosomal membrane acquires LAMPs (lysosome-associated membrane glycoproteins) during the maturation process. The function of these proteins are unknown, but they have been hypothesized to be involved in the protection of the phagosomal membrane from hydrolysis through the contained proteases [85] and in the later stages of endosomal maturation [87]. They are used as markers for late endosomes and phagosomes, and LAMP1 has been observed specifically at *Borrelia*-containing phagosomes (Figure 4) [21].

A close spatial localization between *Borrelia*-containing phagosomes and the endoplasmic reticulum (ER) has been observed during the early stages of the phagosomal maturation process [21,56]. Indeed, stromal interaction molecule 1 (STIM1), a transmembrane component of ER contact sites [88], has been detected at these sites, pointing to the existence of bona fide ER contact sites at *Borrelia* phagosomes [56]. Membrane contact sites (MCS) between the ER and other organelles and structures moved into the focus of ER-related research in recent years. In general, functions ascribed to these MCS include localized calcium signaling, lipid transfer and involvement in membrane trafficking [89]. The specific role of MCS in the processing of *Borrelia* is still unknown. Indeed, MCS at phagosomes have only been described in a limited number of cases [90], which is in contrast to their well-documented role on endosomes [66,91]. Still, several possible functions could be envisioned—most notably, localized calcium signaling. Interestingly, Ca^2+^ levels are increased following activation of FcγR or CR3, both of which are also involved in recognition of *Borrelia*. Furthermore, STIM1 is involved in the pathways during ER-dependent Ca^2+^-signaling [92]. In addition, a role for Ca^2+^-signaling has been reported for the processing of other pathogens in macrophages. Notably, *Mycobacterium tuberculosis* was shown to escape degradation by preventing elevation of Ca^2+^ levels, resulting in defects in acidification and the acquisition of lysosomal enzymes [93]. Further data suggest a role for Ca^2+^ in the regulation of the fusion between phagosomes and lysosomes in a cell-free assay [94]. Collectively, many proteins associated with *Borrelia* phagocytosis are known to be involved in ER-dependent calcium signaling, pointing to possible but currently unproven roles in *Borrelia* intracellular processing.

In addition to regulatory and effector proteins, specific lipids also play important roles in intracellular trafficking and processing of pathogens. For FcγR-mediated phagocytosis of sheep red blood cells (SRBCs) and polysterol beads by RAW 264.7 macrophages, accumulation of PI(3,4,5)P_3_ in the phagosomal cup has been observed [95]. Moreover, inhibition PI(3,4,5)P_3_ formation by targeting PI(3)-kinase class I led to defects in the uptake of SRBCs and latex beads, but did not influence phagosome maturation [96]. Indeed, *Borrelia*-containing phagosomes were also shown to be associated with PI(3,4)P_2_ or PI(3,4,5)P_3_ [60], yet the potential impact of these phospholipids on *Borrelia* phagosome maturation is unclear. In contrast, a clear role has been described for PI(3)P, which was found to become progressively enriched in the membrane of the early phagosome in general [59] and also specifically for *Borrelia* (Figure 4) [60]. As mentioned, it plays an important role in the process of borreliae compaction and recruits Rab5a-/SNX3-positive vesicles to the phagosome by interacting with the PX domain of SNX3 [60]. In line with these findings, pharmacological inhibition of PI(3)P-generating PI(3)-kinase class III by wortmannin disrupted the maturation of latex bead-containing phagosomes in RAW 264.7 macrophages [59,96]. A similar effect for *Borrelia*-containing phagosomes could be envisioned, as Rab5a and SNX3, both present at the vesicles interacting with PI(3)P, were shown to be important for the proteolytic degradation in phagosomes [21,60]. As ER membrane contact sides (MCS) have been described to play a role in lipid transfer, and the composition of the phagosomal membrane changes during development of the phagosome [97], it is conceivable that the mentioned ER-phagosome contacts are also involved in this process.

One of the latest steps in the phagocytic process is the presentation of pathogen-derived peptides in an MHC class II–peptide complex. These complexes are formed within the phagosome and travel from the phagosome back to the cell surface, a process necessary for antigen presentation to other immune cells, and particularly for a targeted T-cell response [98]. Accordingly, upregulation of MHC II has been observed in murine macrophages challenged with *Borrelia* in vivo and during subsequent proliferation of *Borrelia*-reactive T-cells [99]. In contrast, recent data suggest that *Borrelia* might be capable of negatively influencing this signaling, as primary human macrophages challenged with *Borrelia* showed decreased expression of MHCII ex vivo [100]. Further research into this interaction is necessary to evaluate these seemingly discrepant data and understand the processes which might be involved in the pathways for antigen presentation and their potential subversion by borreliae.

## 4. Interactions among Macrophages and Other Immune Cells during *Borrelia* Infection

The immune response to *Borrelia* infection involves many parts of the innate and the adaptive immune system, linked by cytokines and antigen presentation. Cells of the innate immune system are the first line of defense against the spirochete, and include macrophages; polymorphonuclear leukocytes, i.e., neutrophils, eosinophils and basophils; dendritic cells; mast cells; fibroblasts and keratinocytes. Cells of the adaptive immune response consist of CD4+, CD8+, γδ and natural killer (NK) T-cells, and B cells [36].

Dermal fibroblasts are some of the first cells to encounter *Borrelia* during infection. In addition to remodeling of the extracellular matrix, they also play an active role in the context of inflammation by communicating with cells of the immune system and are thus considered to be part of the immune system. When challenged with *Borrelia burgdorferi*, human dermal fibroblasts showed increased release of IL-3, a cytokine which promotes macrophage activation and proliferation, and CCL2, a chemoattractant which draws macrophages to the site of infection (Figure 5) [101,102].

Skin samples from patients with Lyme disease showed elevated levels of IFN-γ and TNFα [102], hinting at the involvement of other cell types in macrophage activation during *Borrelia* infection. IFN-γ is primarily released by T-cells, including NKT cells [103,104,105], whereas TNFα is secreted by macrophages. Both cytokines target and activate macrophages [106].

As antigen presenting cells, dendritic cells and macrophages play crucial roles in the initiation of the adaptive immune response. Following the degradation in the phagolysosome, peptide-MHC II complexes are formed and travel to the cell surface, where they are presented to other immune cells and are especially important for a targeted T-cell response [98,99]. Interestingly, peripheral blood mononuclear cells co-cultured with *Borrelia burgdorferi* expressed and released a range of pro-inflammatory cytokines, mostly in an NF-κB- and MyD88-dependant manner, including IFN-γ, IL12, IL6, IL1, IL8 and TNF-α [107]. The production of type I interferons depends on the phagocytosis of *Borrelia* and on recognition of the pathogen within the phagolysosome by TLR7 and TLR 9, as inhibition of phagocytosis using cytochalasin D or combined inhibition of TLR7 and TLR9 using specific inhibitors abolishes production of IFN-α [107].

## 5. Potential Immune Evasion Strategies of *Borrelia*

Not every infection with *Borrelia* can be completely controlled by the immune system, allowing the development of Lyme disease. In this respect, *Borrelia* could even be recultivated from mouse samples 360 days after the initial inoculation [108]. Comparable evidence for prolonged microbial persistence in humans is lacking, but in some patients, symptoms associated with Lyme disease are known to persist for several months after antibiotic treatment. Whether this is due to microbial persistence, immune dysregulation or other factors is still a matter for debate [16]. In general, mechanisms for immune evasion and escape from phagocytic degradation have been described for different pathogens [29], and *Borrelia burgdorferi* shows a variety of evasion strategies as well.

As previously discussed, borreliae are highly motile, and as such could potentially attempt to escape from the proximity of phagocytic cells. Supporting this hypothesis, membrane tunnels have been observed in primary human macrophages, which extend deeper into the host cytoplasm than the actual *Borrelia*-containing phagosome. Most likely, these structures constitute parts of the nascent phagosome from which the spirochetes manage to extract themselves partially [56]. This dynamic interplay between pathogen and phagocyte would also be in line with the initial requirement for filopodia as pathogen-immobilizing structures, as they help to keep the motile spirochetes near the host cell surface until the more elaborate coiling pseudopod is formed. This seems to be especially important, considering that borreliae can reach velocities of up to 4.25 nm/s [6], whereas macrophages show speeds of up to 10 µm/min [109,110,111], and are thus ~25× slower than their bacterial prey.

Even after phagocytic uptake by macrophages, a small subpopulation of *Borrelia* escapes degradation and survives within the macrophage [86]. Usually, the elongated spirochete is compacted within the phagosome into a globular structure, but in some experiments (1–5% of uptake events), *Borrelia* acquired a phagosomal coat only transiently during uptake and subsequently retained their elongated morphology [21]. The molecular mechanisms involved in this phenomenon are currently unknown. Still, reduced compaction of spirochetes as a result of siRNA-mediated knockdown of phagosomal coat proteins was correlated with increased intracellular survival, as it was possible to recultivate significantly more viable *Borrelia* from lysates of macrophages depleted in Rab5a, Rab22a or SNX3 than from control samples [21,60]. Whether these phenomena are causally linked is yet to be proven, but the observed correlation between compaction and degradation suggests that compaction is likely a necessary step in the processing of the pathogen. Similar results have previously been observed in Vero cells, where a subpopulation of *Borrelia* was not contained within a phagosomal membrane, an effect which was abrogated when using heat-killed *Borrelia*, suggesting active evasion from phagosomal degradation [112]. Other pathogens, such as *Mycobacterium tuberculosis*, are known to target RabGTPases and their effectors or PI(3)P and other components of the phagosomal membrane to arrest phagosomal maturation [29]. Whether similar mechanisms are actively employed by *Borrelia* to escape phagocytosis by macrophages is currently unknown.

Of course, the best defense against phagocytes is to stay out of their way in the first place. As *Borrelia* are transmitted by tick bites, they take advantage of the conditions provided by tick saliva. Tick saliva contains a number of active reagents that lead to vasodilation, supporting increased blood flow and distribution of the pathogen [113]. Moreover, proteins such as Salp20 inhibit the complement system [114], thereby preventing opsonization. In line with this, defects in opsonization were shown to lead to reduced internalization of *Borrelia* by macrophages [42].

A further immune evasion strategy consists of binding the host complement regulator factor H (FH), which includes factor H-binding proteins such as CspA and CspZ, and likely OspE. For more details on complement evasion, the reader is referred to several recent reviews [115,116,117]. In addition, *Borrelia burgdorferi* has been shown to establish a protective niche for itself and evade the humoral immune response by upregulating the decorin binding protein A (DbpA). Of note, DbpA´s interaction partner decorin is highly expressed in the skin and joints, two major sites of infection in the context of Lyme disease. In mice, *Borrelia* loads were increased in these tissues with high decorin expression, promoting symptomatic manifestations, and spirochetes could still be recultivated from joint punctates 15 weeks after infection [118]. *Borrelia* clearance was found to negatively correlate with decorin expression in the tissue and DbpA expression by the spirochete. Furthermore, decorin-deficient mice showed increased pathogen clearance [119].

Collectively, these molecular mechanisms of evading the immune system could support prolonged survival of borreliae within the human host. However, it is currently unclear whether, and if so, to which degree, persisting symptoms reported in some patients are associated with survival of borreliae, or are based on dysregulation of the patients’ immune systems. At the least, these evasion methods certainly have roles to play in the initial confrontation between the immune cells of the host and the pathogen.

## 6. Concluding Remarks and Open Questions

As Lyme disease is the most common tick-borne disease in North America and Europe [2], it is important to understand the underlying pathology. The host immune response against the causative agent, *Borrelia* spirochetes, is a crucial part of the defense against the disease. As professional phagocytes, macrophages play an important part in the attempt to eradicate the pathogen. Moreover, recent research has uncovered important molecular mechanisms involved in the uptake and initial processing of borreliae. The phagocytic process of *Borrelia* thus consists of multiple steps: (I) recognition by surface receptors, and capturing and immobilization by filopodia; (II) internalization by coiling pseudopods; (III) compaction in phagosomes; (IV) degradation in phagolysosomes; and (V) antigen presentation on MHC II complexes. Both dysregulation of individual steps during this process, and potentially also active evasion mechanisms employed by *Borrelia,* could be involved in the pathogenesis of the disease.

Still, the molecular mechanisms behind many steps in the cascade of the intracellular processing of *Borrelia* are only incompletely understood (Figure 6). These include (1) the roles of the ER and phagosomal phospholipids in phagosome maturation, (2) the observed relevance of the compaction process for phagosome maturation, (3) the likely involvement of alternative pathways in this phenomenon that are independent of the described Rab22a–Rab5a–SNX3–galectin-9 axis, (4) the mechanisms regulating membrane tubulation and abscission at phagosomes, (5) the mechanisms involved in loss of the phagosomal coat of a subset of internalized borreliae, (6) the likely involvement of additional regulators and vesicle populations in phagosome maturation, (7) the mechanisms regulating membrane fusion between vesicles and maturing phagolysosomes and (8) the nature of the degradative machinery within mature phagolysosomes.

(1) The observed STIM-1-positive contact sites between the ER and the early stages of the *Borrelia*-containing phagosome should be an interesting field for future research, as little is known about their functions during this process. Multiple roles have been suggested for such contact sites in general, such as local Ca^2+^ signaling and exchange of lipids and proteins [90]. Additionally, changes in the composition of the phagosomal membrane could be important for the maturation of the organelle, comparably to the documented role of PI(3)P in the docking of at least two distinct vesicle populations, leading to phagosome compaction. As other pathogens are known to influence the lipid compositions of membranes as a means to escape immune cells, ER-phagosome contact sites and the composition of the phagosomal membrane should be worth a more detailed investigation.

(2) Uptake and compaction of the highly motile and elongated borreliae requires finely tuned reorganization and restructuring of the cytoskeleton and the phagosomal membrane. Compaction of the spirochete was shown to be a prerequisite for phagosomal maturation and seems to be essential for eventual elimination of the pathogen. However, the reasons for it and the respective molecular mechanisms are only understood in part.

(3) For example, some regulators involved in the pathways leading to phagosomal compaction have already been identified; however, the involvement of other, yet unidentified pathways is likely. 

(4) Furthermore, phagosome compaction is associated with reduction of phagosomal surface, which is based on the formation and abscission of membrane tubules. However, the molecular mechanisms involved in the tubulation and abscission are unknown. 

(5) A subset of borreliae have been observed to retain their elongated form and lose the phagosomal coat [21]. However, it is currently unknown whether this is an active process driven by the pathogen, and if so, which mechanisms are responsible.

The maturation process which transforms the cell membrane-derived nascent phagosome into a phagolysosome capable of destroying the pathogen calls for tightly regulated interaction with other organelles and vesicles. Dysregulation of these processes, either during pathogen uptake or phagosome maturation, whether by a malfunction in a pathway or by active subversion by the pathogen, might lead to increased severity of the disease. It is thus vital to understand both the pathways involved in the regulation of phagocytosis and the potential immune evasion mechanisms employed by *Borrelia* to escape their degradation.

In recent years, multiple proteins have been identified that play pivotal roles in the regulation of *Borrelia* uptake and intracellular processing. Compaction of the phagosome and the trafficking of vesicles to the phagosome, along with their fusion and fission with the phagosomal membrane, are regulated by GTPases of the Rab family, by sorting nexin-3 and by galectin-9. Several RabGTPases have been identified as players in the maturation process of phagosomes of other phagocytic targets. Still, in the case of *Borrelia*, only the particular importance of Rab5a and Rab22a for uptake, compaction and degradation has been demonstrated yet. Of note, other pathogens such as *Mycobacteria* escape degradation by targeting those RabGTPases. The fact that a subset of internalized borreliae lose their Rab22a-positive phagosomal coat could point to the existence of comparable, but so far hypothetical mechanisms for the intracellular survival of *Borrelia*. 

(6,7) The mechanisms involved in phagosomal membrane fusion or contact events are currently only understood in part. Other vesicle populations and regulators than those previously described might yet be identified.

(8) Moreover, little is known about the regulation and relevant enzymes of late-stage phagolysosomes and the eventual degradation of internalized spirochetes. In this regard, cathepsins and NADPH oxidase 2 (NOX2) appear as likely candidates, as they have been observed in lysosomes and phagolysosomes of other phagocytic targets.

Apart from the usual suspects, other players seem to be involved in the processes related to phagocytosis, as new proteins have been found within or in the proximity of phagosomes [120], and functional roles have been documented for some of those proteins. An emerging but so far underappreciated, field concerns the metabolic pathways of the host cell. With the topic of immunometabolism gaining more prominence in recent years, further research in this area, especially in the context of pathogen processing, and specifically *Borrelia* elimination, seems promising.

In summary, closely regulated changes in phagosomal morphology and composition are central for the successful elimination of *Borrelia*. Accordingly, several regulators of intracellular trafficking have already been linked to this process, but many more might be involved. Additionally, borreliae have emerged as being far from passive targets, which actively counteract their capturing and internalization by immune cells. It would thus be highly interesting to see whether borreliae also employ more active mechanisms to influence their intracellular processing and how these could contribute to the pathology of Lyme disease.

## Figures and Tables

**Figure 1 biology-10-00567-f001:**
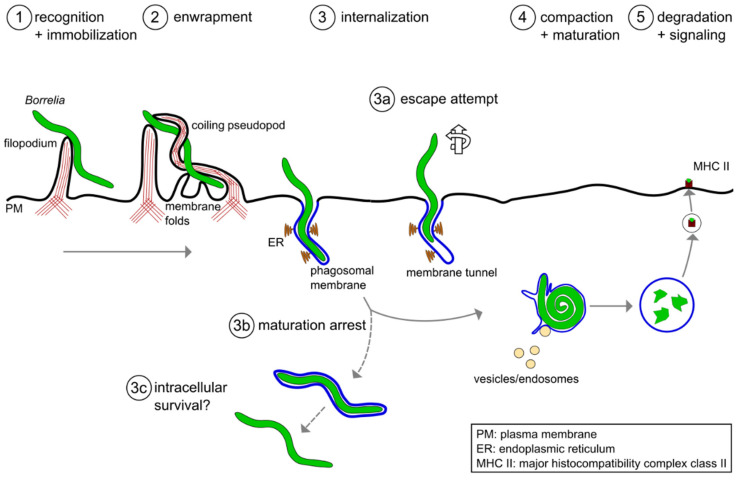
A model of the phagocytic uptake and intracellular processing of *Borrelia* by macrophages. (**1**,**2**) Actin-rich uptake structures. (**1**) Filopodia immobilize motile *Borrelia* at the host cell surface, with (**2**) subsequent enwrapment by a coiling pseudopod. (**3**) Borreliae are taken up into a plasma membrane-derived compartment, the phagosome. The phagosomal membrane loosely follows the spirochete morphology and contacts the endoplasmic reticulum (ER) at multiple sites. (**3a**–**3c**) Potential deviations from the regular pathway of internalization and processing. (**3a**) A subset of borreliae partially extract themselves from the nascent phagosome, resulting in the formation of membrane tunnels. (**3b**) A subpopulation of borreliae lose the phagosomal membrane and retain their elongated morphology, (**3c**) leading to increased survival in the host cell. (**4**) The elongated spirochete is compacted into a globular structure within the phagosome. This is initiated by contact with endocytic vesicles, which leads to local membrane tubulation and thus shrinkage of the phagosomal surface. (**5**) Borreliae are degraded within mature phagolysosomes. Subsequently, complexes consisting of MHCII and pathogen-derived peptides are exposed on the cell surface for antigen presentation.

**Figure 2 biology-10-00567-f002:**
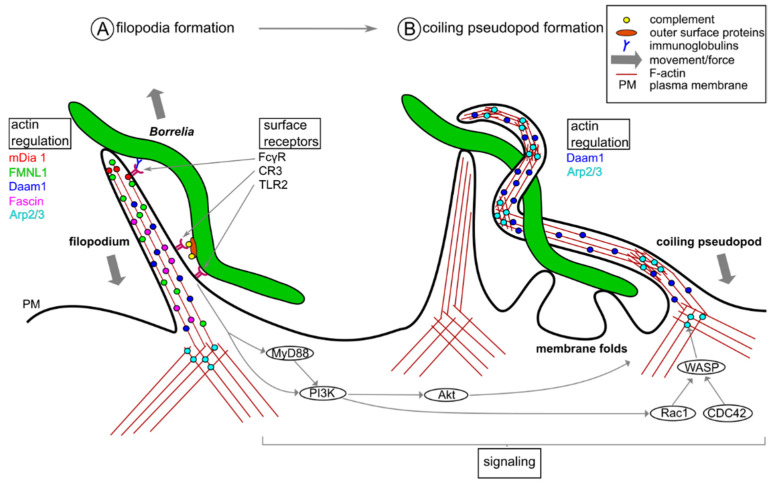
Actin-rich uptake structures during *Borrelia* phagocytosis. (**A**) A filopodium enriched in surface receptors recognizes and attaches to the *Borrelia* cell. The formin mDia1 localizes at the tip of the protrusion, where it regulates actin filament elongation. Two other formins, FMNL1 and Daam1, along with the actin bundling protein fascin, are located along the shaft of the filopodium, and regulate the formation and architecture of unbranched actin filaments. The Arp2/3 complex is found in the underlying branched actin network. The filopodium likely exerts a force to counter movement of the spirochete. (**B**) A second actin-rich structure, the coiling pseudopod, wraps itself around the spirochete, leading to internalization. The Arp2/3 complex is located at the turning points of the pseudopod, likely creating “hinges” of branched F-actin, which alternate with unbranched actin filaments regulated by Daam1. Arp2/3 is activated by a pathway involving Rac1, CDC42 and WASP. Moreover, TLR2-receptor signals in MyD88-dependent and MyD88-independent pathways involving PI3K and Akt, leading to recruitment of Arp2/3, thereby promoting formation of the coiling pseudopod.

**Figure 3 biology-10-00567-f003:**
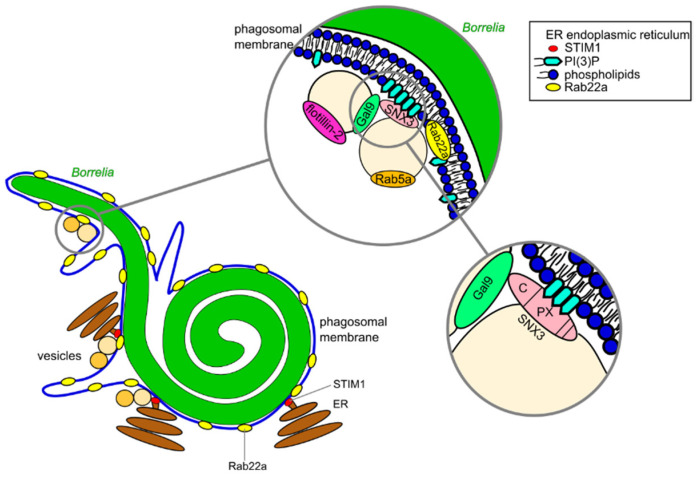
A model of *Borrelia* compaction within the phagosome. The phagosomal membrane is enriched in Rab22a and is met by ER-tethered vesicles positive for both Rab5a and SNX3. Vesicle contact takes place preferentially at sides of altered membrane curvature, owing to the relative enrichment of the phospholipid PI(3)P. The PX domain of SNX3 binds PI(3)P, leading to the docking of Rab5a vesicles, while the SNX3 C-terminus binds galectin-9, resulting in recruitment of a second distinct vesicle population that is positive for flotillin-2. Contact with both vesicle populations is important for the local initiation of membrane tubules, which lead to a reduction of the phagosome’s surface and thus to spirochete compaction.

**Figure 4 biology-10-00567-f004:**
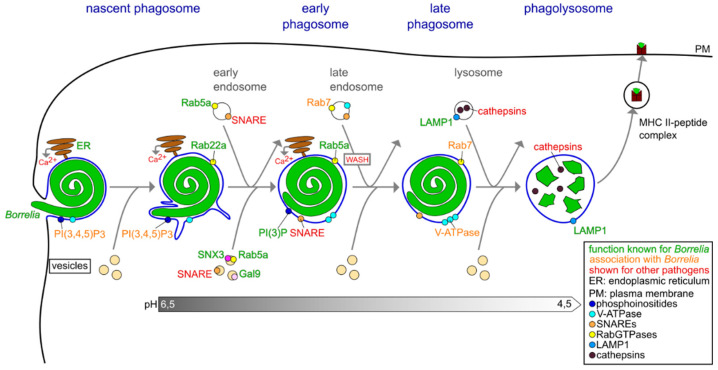
The maturation of *Borrelia*-containing phagosomes. The nascent phagosome makes contact with the ER and is subsequently pinched off from the plasma membrane. During its maturation into a phagolysosome, the compartment comes into contact with different subpopulations of vesicles, which regulate compaction and acidification. This is accompanied by changes in the composition of the phagosomal membrane and also by respective enrichments of different RabGTPases during specific stages. Molecular regulators with documented roles in *Borrelia* phagosome maturation are shown in green, while those that have only been detected, but have unknown roles, are shown in orange. Central regulators of phagosome maturation for other pathogens, with likely but still unproven roles in borreliae phagosome processing, are shown in red.

**Figure 5 biology-10-00567-f005:**
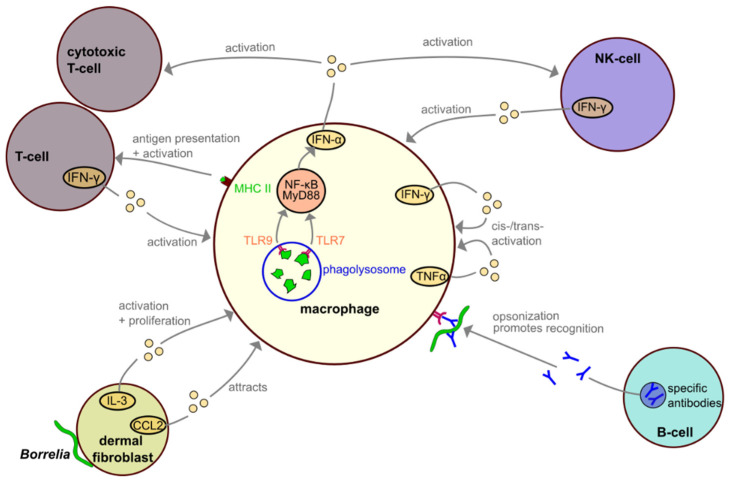
Interactions among macrophages and other immune cells during *Borrelia* infection. Dermal fibroblasts release IL-3 and CCL2 upon encountering *Borrelia*. Activated T-cells and NK-cells, including NKT cells, release IFN-γ. IL-3, CCL2 and IFN-γ activate or attract macrophages. TLR7 and TLR9 recognize *Borrelia* in phagolysosomes and stimulate IFN-α release in NF-κB or MyD88-dependent pathways. Macrophages present *Borrelia* antigens on MHC II-peptide complexes, leading to T-cell activation.

**Figure 6 biology-10-00567-f006:**
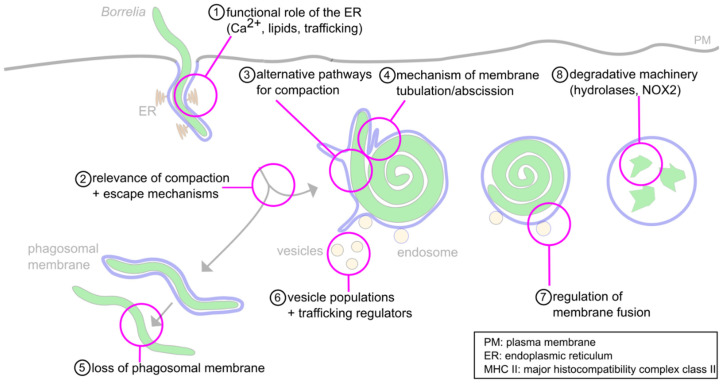
Open questions about the intracellular processing of *Borrelia*. (**1**) The functions of the ER-phagosome contact sites during phagosome compaction and maturation are still unknown. Likely functions concern Ca^2+−^ signaling, lipid transfer and regulation of trafficking. (**2**) The relevance of phagosome compaction for further phagolysosome maturation has been documented. However, the molecular basis for this requirement is unclear. (**3**) While some regulators of compaction have been identified, their individual or combined depletion leads to only a 50% reduction of compaction, indicating the likely involvement of alternative pathways. (**4**) Compaction is accompanied by the extrusion and abscission of membrane tubules; however, the underlying molecular mechanisms are unclear. (**5**) A subpopulation (1–5%) of *Borrelia* lose the phagosomal membrane and retain their elongated morphology. The mechanisms for this process, and its potential importance for survival in the host, are unclear. (**6**) The phagosome is contacted by Rab5a/SNX3 and galectin-9/flotillin-2 vesicles. Involvement of other regulatory cargo proteins and lipids, along with further vesicle populations, is likely but unproven. (**7**) The interaction between vesicles and the phagosomal membrane likely happens in a “kiss-and-run” fashion. The molecular mechanisms are unknown. (**8**) The degradative machinery is well described for other phagocytic targets. However, it is unknown whether the same enzymes, such as cathepsins and NOX-2, are also involved in proteolytic processing of *Borrelia*.

## Data Availability

Not applicable.

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
