# Peer review of "Molecular Mechanisms of Borrelia burgdorferi Phagocytosis and Intracellular Processing by Human Macrophages"

_biology, 2021, doi:10.3390/biology10070567_

Round 1

Reviewer 1 Report

Summary: In their article entitled “Molecular Mechanisms of Borrelia Phagocytosis and Intracellular Processing by Human Macrophages”, Woitzik and Linder review the current knowledge of processes leading to phagocytosis of Borrelia spirochetes by macrophages. I found the review to be well written with a nice focus on molecular mechanisms. I think this is a needed and timely review in the field. I have commented on several areas for improvement, most of which are minor in their nature.

  1. The only major criticism I have that should be resolved is that Borrelia spirochetes are incorrectly referred to as the causative agent of Lyme disease. The genus Borrelia also includes both relapsing fever (RF) spirochetes (i.e. hermsii, B. turicatae, B. recurrentis, etc.) as well as B. miyamotoi, both of which cause infectious diseases in humans that are not related to Lyme disease. All (?) of the literature cited in the review is specific for Lyme disease spirochetes that are now referred to as Borreliella or using the older nomenclature B. burgdorferi sensu lato. There are a large number of places in the manuscript, including the title where this issue will need to be addressed. Alternatively, if studies have been performed using RF species, those should be cited and the delineation made.

  1. The font for the text on ALL figures should be increased. It is very difficult to read most of the labels.

  1. Line 37 states that Borrelia are found in deer. This is not true for Lyme disease spirochetes (see Telford III et. al, 1988, American J. of Trop. Med. And Hyg.). While Ixodes ticks preferentially feed on white-tail deer in the adult stage, the deer themselves are not competent hosts for Borrelia.

  1. The comment about complement receptor 3 (CR3) on lines 106-108 is confusing. While the cited studies do indicate a direct interaction between borrelial proteins and CR3, the canonical ligand for CR3 is the complement cleavage product iC3b. This should probably be mentioned and a study that is relevant that is not cited is Hawley et. al, PNAS, 2012.

  1. Line 470, neutrophiles should be neutrophils.

  1. Line 497 indicates that IL-10 is proinflammatory the way it is written, however, IL-10 is traditionally considered an anti-inflammatory cytokine.

  1. Motility of the spirochete is mentioned a few times in the review, for example in lines 512-520. What are the relative speeds associated with Borrelia cells versus human macrophages? I think this should be mentioned. My understanding was that Borrelia are many times faster and that this difference is thought to contribute significantly to their escape of professional phagocytes.

  1. The consideration of immune evasion on lines 548-562 is insufficient and in the case of complement evasion is not entirely correct. The role of OspE is not well defined in vivo, whereas two other classes of factor H-binding proteins (CspA and CspZ) are. Given that it is not the focus of this review, I suggest making a much more general statement here and referencing a few of the more recent reviews on the topic of complement evasion by Lyme disease spirochetes such as: 1) Dulipati V. et. al FEBS Lett., 2020; 2) Skare JT et. al, Trends in Micro, 2020; 3) Lin YP et. al Front. Cell. Infect. Micro. 2020.

Author Response

Reviewer 1:

In their article entitled “Molecular Mechanisms of Borrelia Phagocytosis and Intracellular Processing by Human Macrophages”, Woitzik and Linder review the current knowledge of processes leading to phagocytosis of Borrelia spirochetes by macrophages. I found the review to be well written with a nice focus on molecular mechanisms. I think this is a needed and timely review in the field. I have commented on several areas for improvement, most of which are minor in their nature.

Thank you for your kind appreciation of our work.

  1. The only major criticism I have that should be resolved is that Borrelia spirochetes are incorrectly referred to as the causative agent of Lyme disease. The genus Borrelia also includes both relapsing fever (RF) spirochetes (i.e. hermsii, B. turicatae, B. recurrentis, ) as well as B. miyamotoi, both of which cause infectious diseases in humans that are not related to Lyme disease. All (?) of the literature cited in the review is specific for Lyme disease spirochetes that are now referred to as Borreliella or using the older nomenclature B. burgdorferi sensu lato. There are a large number of places in the manuscript, including the title where this issue will need to be addressed. Alternatively, if studies have been performed using RF species, those should be cited and the delineation made.

We thank the reviewer for pointing out this important issue. We are now mentioning specifically Borrelia burgdorferi in the title. We also have now added a clarification statement early in the text: “Please note that the term “Borrelia” in this review refers to members of the Borrelia burgdorferi sensu latu complex, especially in the context of Lyme disease, and not to other species such as B. recurrentis or B. miyamotoi that cause other infectious diseases. It is likely that several of the described mechanisms also apply to these other Borrelia species, but this has not been formally proven yet.” (p4, end of Introduction).

  1. The font for the text on ALL figures should be increased. It is very difficult to read most of the labels.

We have now increased font size by 2 points in all figures. For even better legibility, we would kindly ask the editor/production team to increase the size of the figures embedded in the manuscript, as there are still side margins that would allow this.

  1. Line 37 states that Borrelia are found in deer. This is not true for Lyme disease spirochetes (see Telford III et. al, 1988, American J. of Trop. Med. And Hyg.). While Ixodes ticks preferentially feed on white-tail deer in the adult stage, the deer themselves are not competent hosts for Borrelia.

Thank you for pointing this out. “Deer” has now been removed from this sentence.

  1. The comment about complement receptor 3 (CR3) on lines 106-108 is confusing. While the cited studies do indicate a direct interaction between borrelial proteins and CR3, the canonical ligand for CR3 is the complement cleavage product iC3b. This should probably be mentioned and a study that is relevant that is not cited is Hawley et. al, PNAS, 2012.

Thank you for pointing this out. We have now clarified the statement accordingly, including the respective citation provided by the reviewer.

  1. Line 470, neutrophiles should be neutrophils.

Thank you for catching that. It has been corrected throughout the text.

  1. Line 497 indicates that IL-10 is proinflammatory the way it is written, however, IL-10 is traditionally considered an anti-inflammatory cytokine.

Thank you for pointing this out. IL-10 has now been removed from this sentence.

  1. Motility of the spirochete is mentioned a few times in the review, for example in lines 512-520. What are the relative speeds associated with Borrelia cells versus human macrophages? I think this should be mentioned. My understanding was that Borrelia are many times faster and that this difference is thought to contribute significantly to their escape of professional phagocytes.

We have now added numbers for speeds of borreliae (4.25 µm/s) and macrophages (10 µm/min), together with respective citations (point 5), and pointed out the difference between phagocyte and bacterial prey.

  1. The consideration of immune evasion on lines 548-562 is insufficient and in the case of complement evasion is not entirely correct. The role of OspE is not well defined in vivo, whereas two other classes of factor H-binding proteins (CspA and CspZ) are. Given that it is not the focus of this review, I suggest making a much more general statement here and referencing a few of the more recent reviews on the topic of complement evasion by Lyme disease spirochetes such as: 1) Dulipati V. et. al FEBS Lett., 2020; 2) Skare JT et. al, Trends in Micro, 2020; 3) Lin YP et. al Front. Infect. Micro. 2020.

We have now made a more general statement on the role of complement evasion by borreliae, citing the references indicated by the reviewer.

Reviewer 2 Report

This review article by Woitzik and Linder is a nice review of the biology of Borrelia phagocytosis by macrophages, and will be of significant value to the field. This reviewer only has a few minor suggestions to improve the quality of this fine manuscript.

  1. Lines 48, 50, etc. Erythema migrans should not be italicized.
  2. In figure 1, (3c) the process of intracellular survival is very controversial, with only one paper from 1993 showing this (as far as I know). The authors may want to change the solid arrows to dotted arrows to be consistent with the (?) included in the figure, indicating this is not well documented or understood.
  3. Line 126 contains "activated" and "actively" in the same sentence. consider revising.
  4. Line 383, Unpublished observations should not be stated as fact, especially in a review article, prior to peer review. Please revise or omit.
  5. Figure 5 and accompanying text: To be complete, the authors may want to add NKT cells as sources of IFN-gamma. Several papers have published on this cell type.
  6. Lines 495-501. The study cited (103) used human PBMC, not macrophages as stated. Please revise.

Author Response

Reviewer 2:

This review article by Woitzik and Linder is a nice review of the biology of Borrelia phagocytosis by macrophages, and will be of significant value to the field. This reviewer only has a few minor suggestions to improve the quality of this fine manuscript.

Thank you for your kind appreciation of our work.

  1. Lines 48, 50, etc. Erythema migrans should not be italicized.

Thank you for pointing this out. It has now been corrected throughout the text.

  1. In figure 1, (3c) the process of intracellular survival is very controversial, with only one paper from 1993 showing this (as far as I know). The authors may want to change the solid arrows to dotted arrows to be consistent with the (?) included in the figure, indicating this is not well documented or understood.

Thank you for this good suggestion. We have now exchanged the solid arrows for dotted ones.

  1. Line 126 contains "activated" and "actively" in the same sentence. consider revising.

Thank you for this good suggestion. “Actively” has now been removed.

  1. Line 383, Unpublished observations should not be stated as fact, especially in a review article, prior to peer review. Please revise or omit.

We agree with the reviewer and have rephrased this sentence into a more general statement on acidification of phagosomes.

  1. Figure 5 and accompanying text: To be complete, the authors may want to add NKT cells as sources of IFN-gamma. Several papers have published on this cell type.

Thank you for this good suggestion. We have now added NKT cells in the legend to Figure 5 and also in the text of the manuscript, including three respective citations that show their role in Lyme carditis and arthritis. We did not include an additional circular shape for NKT cells in order to keep the figure balanced.

  1. Lines 495-501. The study cited (103) used human PBMC, not macrophages as stated. Please revise.

Thank you for pointing this out. The cell type has now been corrected.